# TEXT2DATA: LOW-RESOURCE DATA GENERATION WITH TEXTUAL CONTROL

## ABSTRACT

Natural language serves as a common and straightforward control signal for humans to interact seamlessly with machines. Recognizing the importance of this interface, the machine learning community is investing considerable effort in generating data that is semantically coherent with textual instructions. While strides have been made in text-to-data generation spanning image editing, audio synthesis, video creation, and beyond, low-resource areas characterized by expensive annotations or complex data structures, like molecules, motion dynamics, and time series, often lack textual labels. This deficiency impedes supervised learning, thereby constraining the application of advanced generative models for text-to-data tasks. In response to these challenges in the low-resource scenario, we propose Text2Data, a novel approach that utilizes unlabeled data to understand the underlying data distribution through an unsupervised diffusion model. Subsequently, it undergoes controllable finetuning via a novel constraint optimization-based learning objective that ensures controllability and effectively counteracts catastrophic forgetting. Comprehensive experiments demonstrate that Text2Data is able to achieve enhanced performance regarding both generation quality and controllability across various modalities, including molecules, motions and time series, when compared to existing baselines.

## 1 INTRODUCTION

Autonomy and controllability stand as twin primary pillars of generative AI (Gozalo-Brizuela & Garrido-Merchan, 2023; Wang et al., 2022). While the challenge of autonomy has been substantially addressed through the rapid advancements of generative models, controllability is now ascending as a fervently explored arena within the machine learning community. As natural languages are one of the most common and simplest control signal for human beings to interact with machines, the machine learning community has increasingly focused on generating data that aligns semantically with textual descriptions, given its wide-ranging applications in fields such as image editing (Zhang et al., 2023; Kawar et al., 2023), audio synthesis (Liu et al., 2023; Huang et al., 2023), video generation (Li et al., 2018; Hu et al., 2022), and many more (Tevet et al., 2023; Sanghi et al., 2022).

Recent breakthroughs in text-to-data generative models, particularly those using diffusion techniques (Li et al., 2023; Yang et al., 2023; Kumari et al., 2023), have demonstrated remarkable proficiency by harnessing the rich semantic insights from vast datasets of data-text pairs. Despite the broad applicability of text-to-data generative models, not all modalities can meet the substantial data-text pair requirements for achieving optimal controllability during model training. This is often due to costly annotations or intricate data structures, a scenario we refer to as the low-resource situation. The lack of text labels in certain areas, such as molecules (Ramakrishnan et al., 2014; Irwin et al., 2012), motions (Guo et al., 2020; Mahmood et al., 2019), and time series (Du et al., 2020), primarily restricts supervised learning and hinders the use of advanced generative models for text-to-data generation tasks. The low-resource situation when training generative models unsurprisingly results in issues like undesirable generation quality, model overfitting, bias, and lack of diversity. As a result, the optimization for scarce text representations to improve the alignment between generated data and input texts in generative models is still under-explored.

To mitigate the issues in the low-resource scenario, the strategies such as data augmentation (Hedderich et al., 2020; Meng et al., 2021), semi-supervised learning (Thomas et al., 2013; Cheuk et al., 2021), and transfer learning (Tits et al., 2020; Yi et al., 2018) are utilized. Yet, each comes across

challenges. Data augmentation, for example, cannot always replicate genuine data fidelity to align accurately with initial text descriptions, and potentially leads to overfitting due to over-reliance on augmented samples. This method also exacerbates the training complexity, intensifying the already high computational demands of diffusion models. For semi-supervised learning, text inherently carries nuances, ambiguities, and multiple meanings. Ensuring that the model maintains the correct interpretation when leveraging unlabeled data is not straightforward. Lastly, while transfer learning offers a solution for limited datasets, it is prone to catastrophic forgetting (Iman et al., 2023), where previous knowledge diminishes as new information (i.e., text descriptions) is introduced.

Alternative to existing solutions, we propose Text2Data, a diffusion-based framework achieving enhanced text-to-data controllability even under low-resource situation. Specially, Text2Data operates in two pivotal stages: **(1) Distribution mastery by leveraging unlabeled data**. This step uses unlabeled data to discern the overarching data distribution via an unsupervised diffusion model, eliminating the semantic ambiguity often associated with semi-supervised approaches. **(2) Controllable finetuning on text-labeled data**. The learned diffusion model is then finetuned by text-labeled data. Distinct from methods reliant on data augmentation, Text2Data abstains from inflating the training dataset and employs cross-attention to further refine data-text alignment. To further enhance this framework, we introduce a novel constraint optimization-based learning objective, aiming to mitigate catastrophic forgetting by regularizing the model parameter space closely to its preliminary space before finetuning. Our contributions are summarized as follows:

- We introduce Text2Data, a novel framework designed for text-to-data generation in the low-resource scenario. This approach is able to maintain the fine-grained data distribution by fully harnessing both labeled and unlabeled data.

- We design a novel learning objective based on constraint optimization to achieve controllability and overcome catastrophic forgetting during finetuning, ensuring the model parameter space remains aligned with its initial training phase.

- We theoretically validate our optimization constraint selection and established generalization bounds for our learning objective.

- We compile real-world datasets across three modalities and conduct comprehensive experiments to show the effectiveness of Text2Data. The results demonstrate that Text2Data achieves superior performance than baselines regarding both generation quality and controllability.

## 2 RELATED WORKS

### 2.1 TEXT-TO-DATA GENERATIVE DIFFUSION

Diffusion models, notably divided into classifier-guided (Dhariwal & Nichol, 2021) and classifier-free (Ho & Salimans, 2022) categories, have significantly impacted data generation across various domains. (Hoogeboom et al., 2022; Yang et al., 2023; Ho et al., 2022; Voleti et al., 2022). The classifier-guided diffusion guides the model during inference phase to generate data with desired properties by independently training a classifier and supervising the model with its gradient, which is not efficient when computing gradient at each time step and sometimes the generation quality is deficient as the guidance is not involved in the training. By contrast, classifier-free diffusion guidance blends score estimates from both a conditional diffusion model and an unconditional one with time step as a parameter, exemplified by E(3) Equivariant Diffusion Model (EDM) (Hoogeboom et al., 2022) and Motion Diffusion Model (MDM) (Tevet et al., 2023) for controllable molecule and motion generation, respectively. Furthermore, since natural languages are a prevalent medium for human to communicate with the world, the text-to-data generation paradigm has gained traction, with diffusion models being instrumental in generating high-quality data aligned with textual inputs. The extensive applications encompass text-to-image generation (Ruiz et al., 2023; Zhang & Agrawala, 2023), text-to-speech generation (Huang et al., 2022; Kim et al., 2022), text-to-shape generation (Li et al., 2023; Lin et al., 2023), and more, leveraging the abundant text descriptions for training potent generative models. Various modalities may not satisfy the stringent requirements for data-text pairs essential for attaining optimal controllability during the training of models.

### 2.2 MACHINE LEARNING MODEL UNDER LOW-RESOURCE SITUATION

In response to the challenges of low-resource training for controllable generative models, several strategies have been formulated. For instance, Yin et al. (2023) proposes Text-to-Text-to-Image Data

Augmentation that employed both large-scale pretrained Text-to-Text and Text-to-Image generative models for data augmentation to generate photo-realistic labeled images in a controllable manner. Zang & Wan (2019) utilizes a semi-supervised approach to augment the training of both the encoder and decoder with both labeled and unlabeled data. Tu et al. (2019) proposes to learn a mapping between source and target linguistic symbols and employs transfer learning to transfer knowledge from a high-resource language to low-resource language. Nevertheless, all those strategies have their own limitations, such as high computational complexity for data augmentation, difficulty in maintaining the correct interpretation of text when leveraging unlabeled data during semi-supervised learning, and the potential catastrophic forgetting issues in transfer learning. Additionally, most of works on low-resource text-to-data generation lie in the modality of image, speeches, and texts, which have plenty of datasets (Ito & Johnson, 2017; Pratap et al., 2020; Lin et al., 2014; Jiang et al., 2021; Wang et al., 2020) to train the models nowadays. Yet it is far under-explored for the other low-source modalities such as molecules, motions and time series. Therefore, we propose Text2Data, a diffusion-based framework adept at harnessing limited text-labeled data to enhance the controllability of the model in text-to-data generation.

## 3 PROBLEM FORMULATION

Suppose the dataset $\mathcal{D} = \{\mathbf{x}, \mathbf{c}\}$ contains $N$ independent samples in total, where $\mathbf{x} = \{\mathbf{x}_i\}_{i=1}^{N}$ is the data samples such as molecules, motions, time series, etc. We assume that there is only a proportion of data in $\mathbf{x}$ that has corresponding text description $\mathbf{c} = \{\mathbf{c}_i\}_{i=1}^{N_p}$ where $N_p \leq N$. We denote that data with text description is contained in $\mathcal{D}_p$ and $\mathcal{D}_p \in \mathcal{D}$. Using both text-labeled and unlabeled data in $\mathcal{D}$, we aim to learn a generative model, $p_\theta(\mathbf{x}|\mathbf{c})$ parameterized by $\theta$ that is able to generate data $\mathbf{x} \sim p_\theta(\mathbf{x}|\mathbf{c})$ corresponding to specific text description $\mathbf{c} = \mathbf{c}^*$.

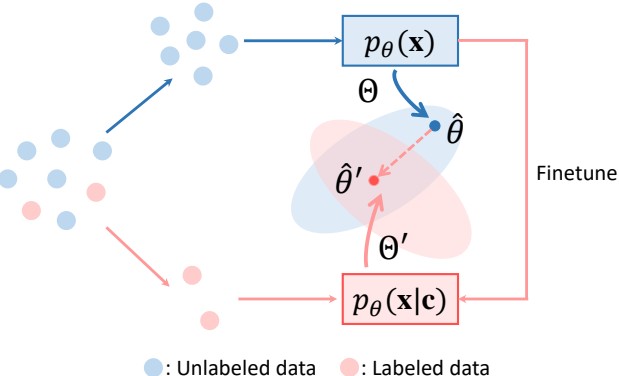

: Unlabeled data  : Labeled data

Figure 1: Overview of Text2Data. The model first leverages unlabeled data (i.e., blue module) to discern the general data distribution and the optimal set of model parameters $\Theta$ is obtained. Then the model is finetuned on labeled data (i.e., red module) by constraint optimization that introduces the optimal set of model parameters as $\Theta \cap \Theta'$, where $\Theta'$ is the optimal set of parameters if finetune the model without constraint.

## 4 METHODS

Controllable data generation seeks to learn the conditional data distribution $p_\theta(\mathbf{x}|\mathbf{c})$ during training and subsequently draw samples from this assimilated distribution during the inference stage. Consequently, our primary objective during the training phase is to optimize the following:

$$\min_\theta \ \mathbb{E}_{\mathbf{x},\mathbf{c} \sim p_{\mathcal{D}_p}(\mathbf{x},\mathbf{c})}[-\log p_\theta(\mathbf{x}|\mathbf{c})].  \quad (1)$$

While the training of such generative models is contingent upon the supervision of text descriptions present in the dataset (denoted as $\mathcal{D}_p$), it is not always feasible to obtain an adequate number of data-text pairs to ensure optimal controllability (i.e., $|\mathcal{D}_p| < |\mathcal{D}|$), especially in specific modalities like molecular structures, motion patterns and time series. Such constraints can precipitate complications, including model overfitting when optimizing according to Eq. 1. Given these challenges, devising strategies to capitalize on the unlabeled data within $\mathcal{D}$—which is often more accessible and cost-effective—is pivotal to effectively learn $p_\theta(\mathbf{x}|\mathbf{c})$.

Notably, the marginal distributions learned from unlabeled and labeled data are close:

$$p_\theta(\mathbf{x}) \approx \int p_\theta(\mathbf{x}|\mathbf{c})p_{\mathcal{D}_p}(\mathbf{c})d\mathbf{c} = \mathbb{E}_{\mathbf{c}\sim p_{\mathcal{D}_p}(\mathbf{c})}[p_\theta(\mathbf{x}|\mathbf{c})], \tag{2}$$

where $p_{\mathcal{D}_p}(\mathbf{c})$ is the true underlying text generating distribution corresponding to $\mathbf{x} \in \mathcal{D}_p$. Hence, as per Figure 1, we can initially utilize the unlabeled data in $\mathcal{D}$ to learn $p_\theta(\mathbf{x})$ and obtain the optimal set of model parameters $\hat{\theta} \in \Theta$, which serves as a robust approximation of $p_\theta(\mathbf{x}|\mathbf{c})$. Then, we finetune it using the data-text pairs in $\mathcal{D}_p$ to achieve desired model controllability (Section 4.1). Crucially, we ensure that the parameters remain in close proximity to those established when learning $p_\theta(\mathbf{x})$ via constraint optimization to make Eq. 2 hold to mitigate catastrophic forgetting. Figure 1 shows our constraint to keep the finetuned parameters $\hat{\theta}'$ within $\Theta \cap \Theta'$, where $\Theta'$ represents the optimal parameters from unconstrained finetuning. Then we plug the generative diffusion implementation to our framework in Section C. Finally, we use Section A to offer theoretical insights into the selection of the optimization constraint, underpinned by generalization bounds for our learning objective.

### 4.1 LEARNING TEXT-TO-DATA GENERATION UNDER LOW-RESOURCE SITUATION

To capture a general distribution of data, we leverage all data in $\mathcal{D}$ incorporating NULL tokens as conditions to facilitate subsequent finetuning. Specifically, we train the generative model $p_\theta(\mathbf{x}|\emptyset)$, where $\theta$ parameterizes the model and $\emptyset$ represents the NULL token in practice. As the NULL token is independent to $\mathbf{x}$, we also have $p_\theta(\mathbf{x}|\emptyset) = p_\theta(\mathbf{x})$. Hence, we optimize the following:

$$\min_\theta \ \mathbb{E}_{\mathbf{x}\sim p_{\mathcal{D}}(\mathbf{x})}[-\log p_\theta(\mathbf{x})], \tag{3}$$

where $p_{\mathcal{D}}(\mathbf{x})$ is the true underlying data generating distribution.

Having discerned the general data distribution through Eq. 3, we proceed to finetune $p_\theta(\mathbf{x}|\mathbf{c})$ using text labels $\mathbf{c}$ in $\mathcal{D}_p$ to achieve the controllability of model. In alignment with Eq. 2, the anticipated finetuned parameter should approximate the parameter optimized in Eq. 3. This leads to the subsequent learning objective for the finetuning phase:

$$\min_\theta \ \mathbb{E}_{\mathbf{x},\mathbf{c}\sim p_{\mathcal{D}_p}(\mathbf{x},\mathbf{c})}[-\log p_\theta(\mathbf{x}|\mathbf{c})]$$
$$\text{s.t. } \mathbb{E}_{\mathbf{x}\sim p_{\mathcal{D}_p}(\mathbf{x})}[-\log p_\theta(\mathbf{x})] \le \xi, \ \xi = \inf_{\theta\in\Theta} \mathbb{E}_{\mathbf{x}\sim p_{\mathcal{D}}(\mathbf{x})}[-\log p_\theta(\mathbf{x})] \tag{4}$$

where $p_{\mathcal{D}_p}(\mathbf{x},\mathbf{c})$ is the true underlying data-text joint distribution. $\Theta$ denotes a localized parameter space where a minimum can be located. Specifically, we minimize $\mathbb{E}_{\mathbf{x},\mathbf{c}\sim p_{\mathcal{D}_p}(\mathbf{x},\mathbf{c})}[-\log p_\theta(\mathbf{x}|\mathbf{c})]$ using the labeled data in $\mathcal{D}_p$ within the optimal set $\{\theta : \mathbb{E}_{\mathbf{x}\sim p_{\mathcal{D}_p}(\mathbf{x})}[-\log p_\theta(\mathbf{x})] \le \xi\}$ to make the parameter not far from learned via Eq. 3, so that catastrophic forgetting is mitigated. This aligns with the nature of a *lexicographic optimization* problem (Gong & Liu, 2021) and therefore can be solved in that context.

### 4.2 LOW-RESOURCE GENERATIVE OBJECTIVE ON EMPIRICAL SAMPLES

Eq. 4 is the population-level learning objective while empirically we follow the standard loss function (i.e., transformed evidence lower bound of Eq. 4) of the classifier-free diffusion guidance (Ho & Salimans, 2022), so that instead of optimizing Eq. 4, we optimize the following:

$$\min_\theta \ \mathcal{L}_2(\theta) \quad \text{s.t. } \mathcal{L}_1'(\theta) \le \xi, \ \xi = \inf_{\theta\in\Theta} \mathcal{L}_1(\theta), \tag{5}$$

where $\mathcal{L}_1(\theta) = \mathbb{E}_{\mathbf{x}\sim p_{\mathcal{D}}(\mathbf{x}),t}[||\epsilon_\theta(\mathbf{x}^{(t)},t) - \epsilon||^2]$, $\mathcal{L}_1'(\theta) = \mathbb{E}_{\mathbf{x}\sim p_{\mathcal{D}_p}(\mathbf{x}),t}[||\epsilon_\theta(\mathbf{x}^{(t)},t) - \epsilon||^2]$ and $\mathcal{L}_2(\theta) = \mathbb{E}_{\mathbf{x},\mathbf{c}\sim p_{\mathcal{D}_p}(\mathbf{x},\mathbf{c}),t}[||\epsilon_\theta(\mathbf{x}^{(t)},\mathbf{c},t) - \epsilon||^2]$. Also, $t$ is sampled from uniform between 1 and $T$, $T$ is the total number of diffusion steps, $\epsilon$ is the standard Gaussian random variable, and $\epsilon_\theta(\mathbf{x}_i^{(t)},t)$ and $\epsilon_\theta(\mathbf{x}_i^{(t)},\mathbf{c},t)$ are functions we aim to fit at the $t$-th diffusion step. Note here that $\epsilon_\theta(\mathbf{x}_i^{(t)},t)$ and $\epsilon_\theta(\mathbf{x}_i^{(t)},\mathbf{c},t)$ share the same parameter but are just trained at different stages: distribution mastery on unlabeled data and controllable finetuning on labeled data, respectively.

As the true generating distributions $p_{\mathcal{D}}(\mathbf{x})$, $p_{\mathcal{D}_p}(\mathbf{x})$ and $p_{\mathcal{D}_p}(\mathbf{x},\mathbf{c})$, are unknown, we instead optimize the empirical loss:

$$\min_\theta \ \hat{\mathcal{L}}_2(\theta) \quad \text{s.t. } \hat{\mathcal{L}}_1'(\theta) \le \hat{\xi}, \ \hat{\xi} = \inf_{\theta\in\hat{\Theta}} \hat{\mathcal{L}}_1(\theta), \tag{6}$$

where $\hat{\mathcal{L}}_1(\theta) = \mathbb{E}_{\mathbf{x} \sim \hat{p}_{\mathcal{D}}(\mathbf{x}), t}[||\boldsymbol{\epsilon}_\theta(\mathbf{x}^{(t)}, t) - \boldsymbol{\epsilon}||^2]$, $\hat{\mathcal{L}}_1'(\theta) = \mathbb{E}_{\mathbf{x} \sim \hat{p}_{\mathcal{D}_p}(\mathbf{x}), t}[||\boldsymbol{\epsilon}_\theta(\mathbf{x}^{(t)}, t) - \boldsymbol{\epsilon}||^2]$ and $\hat{\mathcal{L}}_2(\theta) = \mathbb{E}_{\mathbf{x}, \mathbf{c} \sim \hat{p}_{\mathcal{D}_p}(\mathbf{x}, \mathbf{c}), t}[||\boldsymbol{\epsilon}_\theta(\mathbf{x}^{(t)}, \mathbf{c}, t) - \boldsymbol{\epsilon}||^2]$. $\hat{\Theta}$ is the localized parameter space where a minimum can be located for $\hat{\mathcal{L}}_1(\theta)$. The lexicographic optimization-based constraint $\hat{\xi} = \inf_{\theta \in \hat{\Theta}} \hat{\mathcal{L}}_1(\theta)$ in Eq. 6 may be overly strict and could require relaxation to ease the training process. While we anticipate that the parameters derived from Eq. 6 should be close to those from Eq. 5, they do not necessarily have to be an exact subset of the parameters from Eq. 5.

### 4.3 GENERALIZATION BOUND OF LEARNING CONSTRAINT

In this section, we deduce a confidence bound for the constraint to demonstrate that the optimal set of minimizing the empirical loss within the derived confidence bound encapsulates the true one, and guide the further relaxation on the constraint if needed. First, we define sub-exponential random variable and then and elucidate its relationship with sub-Gaussian random variable.

**Definition 1** (Sub-Gaussian random variable). The random variable $X$ with mean 0 is sub-Gaussian with variance $\sigma^2$ if $\forall s \in \mathbb{R}$, $\mathbb{E}_X[\exp(sX)] \leq \exp(\frac{\sigma^2 s^2}{2})$.

Based on the fact that $\mathbf{x}^{(t)}$ is diffused from the random variable $\mathbf{x}^{(0)}$ and the standard Gaussian noise $\boldsymbol{\epsilon}$, therefore, $\boldsymbol{\epsilon}_\theta(\mathbf{x}^{(t)}, t)$ and $\boldsymbol{\epsilon}_\theta(\mathbf{x}^{(t)}, \mathbf{c}^{(t)}, t)$ are also random variables. Meanwhile, minimizing the loss in Eq. 5 pushes $\boldsymbol{\epsilon}_\theta(\mathbf{x}^{(t)}, t)$ and $\boldsymbol{\epsilon}_\theta(\mathbf{x}^{(t)}, \mathbf{c}^{(t)}, t)$ towards $\boldsymbol{\epsilon}$, which has the mean zero. Then we introduce the following theorem 1 while assuming $\boldsymbol{\epsilon}_\theta(\mathbf{x}^{(t)}, t)$ and $\boldsymbol{\epsilon}_\theta(\mathbf{x}^{(t)}, \mathbf{c}^{(t)}, t)$ are sub-Gaussian random variables.

**Theorem 1.** *For every $\theta$ and $t$, assume $\boldsymbol{\epsilon}_\theta(\mathbf{x}^{(t)}, t)$ and $\boldsymbol{\epsilon}_\theta(\mathbf{x}^{(t)}, \mathbf{c}_i, t)$ are sub-Gaussian random variables with mean 0 and variance $\sigma^2$, and $\Theta$ is finite. Let $\Theta^* = \{\theta : \mathcal{L}_1'(\theta) \leq \xi\}$, $\hat{\Theta}^* = \{\theta : \hat{\mathcal{L}}_1'(\theta) \leq \hat{\xi} + \epsilon\}$ where $\epsilon$ is the confidence bound with the probability of $1 - \delta$. Let $\theta^*$ denote the solution to Eq. 5 and $\hat{\theta}^*$ denote the solution to empirical Eq. 6, then we have the following:*

1. *$\Theta^* \subseteq \hat{\Theta}^*$: the set of $\theta$ by optimizing Eq. 6 within the confidence bound covers the true one.*

2. *$\mathcal{L}_2(\hat{\theta}^*) \leq \mathcal{L}_2(\theta^*) + 2\epsilon_{N_p}$: $\theta^*$ and $\hat{\theta}^*$ compete well on $\mathcal{L}_2(\theta)$.*

3. *$\mathcal{L}_1'(\hat{\theta}^*) \leq \xi + 2\epsilon_{N_p} + 2\epsilon_N$: $\hat{\theta}^*$ does not violate constraint of Eq. 5 too much on $\mathcal{L}_1'(\theta)$.*

*where $\epsilon = \epsilon_N + \epsilon_{N_p}$, $\epsilon_N = \sqrt{C\tilde{\sigma}^2} \cdot \sqrt{\frac{\log |\Theta| + \log \frac{2}{\delta}}{N}} \vee C\tilde{\sigma}^2 \cdot \frac{\log |\Theta| + \log \frac{2}{\delta}}{N}$, $\epsilon_{N_p} = \sqrt{C\tilde{\sigma}^2} \cdot \sqrt{\frac{\log |\Theta| + \log \frac{2}{\delta}}{N_p}} \vee C\tilde{\sigma}^2 \cdot \frac{\log |\Theta| + \log \frac{2}{\delta}}{N_p}$, $\tilde{\sigma}^2 = \sigma^2 + 1$ and $C = 8\sqrt{2}$.*

Theorem 4.3 can be proved starting from the Bernstein's inequality and the union bound inequality on squared zero-mean sub-Gaussian random variable. More detailed proof is in Appendix A. As the large amount of unlabeled data (i.e., $N$) is usually easy to obtain by either manual collection or simulation, $\epsilon_N$ is not large after taking logarithm on the number of model parameters (i.e., $|\Theta|$), even though it is usually much larger than $N$. Additionally, $\log |\Theta|$ is not significantly larger than $N_p$ so that $\epsilon_{N_p}$ should not be large as well. For instance, in our experiments, around 45,000 samples and 14 million model parameters for motion generation result in rather small $\epsilon_N$ and $\epsilon_{N_p}$. In practice, we use $\xi = \rho \cdot \inf_{\theta \in \Theta} \mathcal{L}_1(\theta)$ in Eq. 5 (i.e., $\hat{\xi} = \rho \cdot \inf_{\theta \in \hat{\Theta}} \hat{\mathcal{L}}_1(\theta)$ in Eq. 6) to relax the constraint, where $\rho$ is an hyperparameter to keep the constraint within the confidence interval. Detail learning algorithm has been displayed in Alg. 1.

## 5 EXPERIMENTS

### 5.1 DATASETS

We employ datasets from three modalities that may suffer from low-resource scenario.

**Molecules**. We extract 130,831 molecules from QM9 dataset (Ramakrishnan et al., 2014) with six molecular properties: polarizability ($\alpha$), highest occupied molecular orbital energy ($\epsilon_{\text{HOMO}}$), lowest unoccupied molecular orbital energy ($\epsilon_{\text{LUMO}}$), the energy difference between HOMO and LUMO ($\Delta_\epsilon$), dipole moment ($\mu$) and heat capacity at 298.15K ($C_v$).

**Motions**. We employ HumanML3D that contains textually re-annotating motions captured from the AMASS (Mahmood et al., 2019) and the HumanAct12 (Guo et al., 2020). It contains 14,616 motions annotated by 44,970 textual descriptions.

**Time Series**. We assemble 24 stocks from Yahoo Finance[1] (Appendix B) during their IPO date to July 8, 2023. We further tailor data with the length of 120 by slicing on the opening price for every 120 days, and scale the data by min-max normalization following Yoon et al. (2019). In total, 210,964 time series are produced. We then extract their features including frequency, skewness, mean, variance, linearity (measured by $R^2$), and number of peaks via "tsfresh" in Python.

Each dataset is divided into training and testing sets at a ratio of $80\%$ to $20\%$. We curate each dataset to have varying proportions (i.e., $2\%, 4\%, 6\%, 8\%, 10\%, 20\%, 30\%, 40\%$) of text labels in order to assess Text2Data and its baseline comparisons. Details regarding constructing text descriptions for Molecules and Time Series are introduced in Appendix B.

## 5.2 BASELINE MODELS

We compare Text2Data with a classifier-free diffusion model in each modality as the baselines.

**E(3) Equivariant Diffusion Model (EDM)** (Hoogeboom et al., 2022).To handle Molecule dataset, we employ EDM as the baseline. EDM utilizes an equivariant network to denoise diffusion processes by concurrently processing both continuous (atom coordinates) and categorical data (atom types). The controllability of molecular properties are realized by the classifier-free diffusion guidance conditioned on the embedding of text descriptions and the number of atoms.

**Motion Diffusion Model (MDM)** (Tevet et al., 2023). We use MDM, a classifier-free diffusion model, for text-to-human motion generation. The text descriptions are embedded to guide the motions, providing a mechanism for controllability.

**Generation diffusion for time series (DiffTS)**. To generate time series, we design the classifier-free diffusion model from scratch conditioned on text embeddings. We employ the backbone of Ho & Salimans (2022) by substituting image data to one-dimensional time series, and replacing the U-Net with one-dimensional convolutional neural network.

During implementation, we train the baselines on a specific proportion of labeled data. Text2Data is modified from the baseline model with a pretraining+finetuning strategy following Eq. 6. Additionally, we conduct ablation study where each baseline is still finetuned but without the constraint in Eq. 6. More details are in Appendix C.

## 5.3 EVALUATION METRICS

We evaluate Text2Data and the baselines according to (1) generation quality and (2) controllability.

**Generation quality**. The evaluation of generation quality varies based on the modality. For molecular generation, we compute $-\log$ *likelihood* $(-\log p)$ (Hoogeboom et al., 2022) and *validity* of generated molecules (Jin et al., 2018), *molecular stability* and *atom stability* (Garcia Satorras et al., 2021) to evaluate their overall generation quality. For motion generation, we use *FID score* and *Diversity* following (Tevet et al., 2023). For time series generation, we follow Yoon et al. (2019) and Lim et al. (2023) by drawing t-SNE plots to visualize the overlap between generated data and the ground-truth. Better model tends to have a larger overlap, indicating more similar distribution.

**Controllability**. We compare the controllability between Text2Data and the baselines by the similarity between generated data and the ground truth. To assess the generated molecules, we follow Hoogeboom et al. (2022) and train a classifier for each property to extract specific properties from generated data. Then, we calculate the *Mean Absolute Error* (MAE) between the extracted property and the ground truth. To assess the controllability of motion generation, we compute *R precision* and *Multimodal distance* that measure the relevancy of the generated motions to the input prompts (Guo et al., 2022b). To evaluate the controllability of time series generation, we extract properties via "tsfresh" and compute the MAE between properties of generated data and that of ground truth. Additionally, we also visualize generated data according to the specified properties.

---

[1]https://finance.yahoo.com/

## 5.4 COMPARISONS ON GENERATION QUALITY

When generating molecules from Text2Data and its baseline comparisons, as shown in Table 1, we compute $-\log p$ and validity to evaluate generation quality. The performance of Text2Data is consistently better. It surpasses EDM-finetune and EDM by average margins of 19.07% and 58.03%, respectively. It is 1.98% and 10.59% better than EDM-finetune and EDM on average, respectively, regarding validity on average. Besides, we evaluate Text2Data compared with EDM-finetune and EDM on molecular stability and atom stability (Appendix Table 5). Text2Data exceeds EDM-finetune and EDM by average margin of 2.34% and 17.31%, respectively, in terms of molecular stability. It is also 0.29% and 1.21% better than EDM-finetune and EDM on average, respectively, regarding atom stability. The consistent improvements on all the three models result from our superior performance of Text2Data on properties (e.g., molecular stability) that are hard to control.

Table 1: Evaluate generation quality on Molecule dataset by $-\log p$ and validity according to different proportions of paired data. Lower $-\log p$ and higher validity indicate better performance.

| Proportion (%) | -log p ↓ | | | Validity ↑ | | |
|---|---|---|---|---|---|---|
| | Text2Data | EDM-finetune | EDM | Text2Data | EDM-finetune | EDM |
| 2 | **-111.39±0.92** | -74.88±1.82 | -49.15±0.96 | **0.97±0.07** | 0.93±0.11 | 0.86±0.09 |
| 4 | **-119.09±0.30** | -87.56±4.18 | -78.72±2.95 | 0.95±0.07 | **0.96±0.07** | 0.83±0.15 |
| 6 | **-119.55±0.59** | -97.32±2.09 | -69.58±1.97 | **0.97±0.07** | 0.96±0.05 | 0.81±0.14 |
| 8 | **-119.40±0.68** | -101.31±1.11 | -85.19±1.58 | **0.97±0.07** | 0.93±0.07 | 0.90±0.11 |
| 10 | **-121.37±1.24** | -104.17±1.94 | -85.73±1.00 | **0.96±0.06** | 0.95±0.13 | 0.88±0.10 |
| 20 | **-119.58±1.61** | -104.08±2.03 | -76.39±2.11 | **0.97±0.07** | 0.95±0.07 | 0.90±0.09 |
| 30 | **-121.00±1.07** | -115.58±0.95 | -76.22±1.26 | **0.97±0.07** | 0.95±0.07 | 0.91±0.09 |
| 40 | **-119.90±0.97** | -114.00±0.59 | -80.97±0.09 | **0.97±0.07** | 0.95±0.08 | 0.90±0.10 |

Table 2: Evaluate generation quality on HumanML3D dataset by FID and Diversity according to different proportions of paired data. Low FID and higher diversity indicate better performance.

| Proportion (%) | FID ↓ | | | Diversity ↑ | | |
|---|---|---|---|---|---|---|
| | Text2Data | MDM-finetune | MDM | Text2Data | MDM-finetune | MDM |
| 2 | **1.22±0.12** | 1.23±0.03 | 3.09±0.12 | **9.26±0.07** | 9.08±0.17 | 8.84±0.14 |
| 4 | **1.12±0.11** | 1.16±0.17 | 3.06±0.21 | **9.31±0.21** | 9.13±0.20 | 8.80±0.18 |
| 6 | 1.00±0.13 | **0.64±0.11** | 1.13±0.12 | **9.40±0.13** | 9.39±0.22 | 8.91±0.22 |
| 8 | 1.40±0.14 | **1.32±0.15** | 1.34±0.18 | **9.43±0.20** | 9.21±0.18 | 8.95±0.19 |
| 10 | **1.49±0.19** | 1.52±0.13 | 1.50±0.15 | 9.59±0.09 | **9.74±0.13** | 9.37±0.15 |
| 20 | **0.92±0.06** | 1.02±0.12 | 1.07±0.06 | **9.77±0.20** | 9.72±0.17 | 9.66±0.15 |
| 30 | **0.81±0.10** | 0.99±0.13 | 1.11±0.10 | **9.79±0.11** | 9.70±0.12 | 9.63±0.14 |
| 40 | **0.63±0.12** | 0.95±0.11 | 1.13±0.15 | **9.74±0.13** | 9.70±0.20 | 9.38±0.15 |

As indicated in Table 2, quantitative assessment of motion generation from text shows that Text2Data surpasses the baseline methods in both quality and diversity. Particularly, Text2Data outperforms MDM-finetune and MDM by 2.73% and 36.05% on average, respectively, regarding FID score. Regarding diversity, Text2Data surpasses MDM-finetune and MDM by average margins of 0.81% and 3.71%, respectively. Enhanced performance is derived from the ability of Text2Data to fully leverage all samples in the dataset, while effectively mitigating catastrophic forgetting during finetuning.

We evaluate the overall quality of generating time series by making t-SNE plots of generated time series against ground truth. Substantial overlap between the generated time series and the ground truth suggests a closer distribution alignment, implying a better performance. As demonstrated in Figure 2, the red pattern represents the t-SNE of ground-truth time series, whereas the blue pattern is the t-SNE of generated time series according to the same text description. Compared with DiffTS-finetune and DiffTS, Text2Data corresponds to the largest overlap between distributions of the generated and the ground-truth time series, suggesting its superior ability to precisely generate data according to the text description. The non-overlapping part may result from the diversity of generated time series or properties that are not controlled by text description. The inferior performance of DiffTS stems from its training solely on labeled data, potentially leading to an incomplete understanding of the overall data distribution and a risk of overfitting. DiffTS may only partially capture the data distribution based on textual descriptions due to its susceptibility to catastrophic forgetting, which also heightens the risk of overfitting.

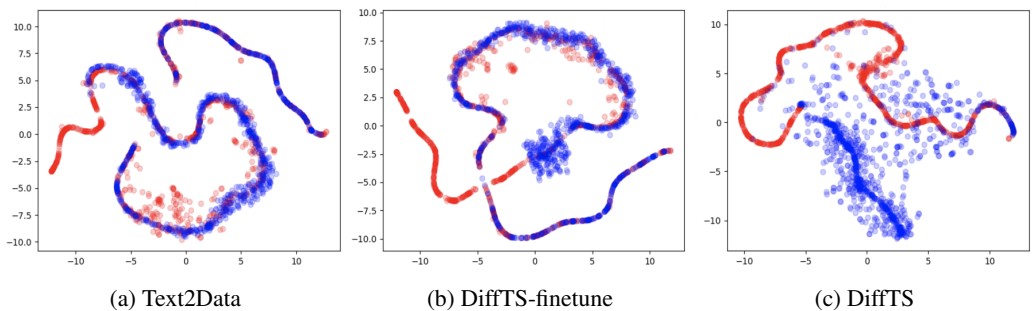

(a) Text2Data      (b) DiffTS-finetune      (c) DiffTS

Figure 2: t-SNE visualization on time series data generated by Text2Data, DiffTS-finetune model and DiffTS. Red denotes ground truth, and blue denotes generated data.

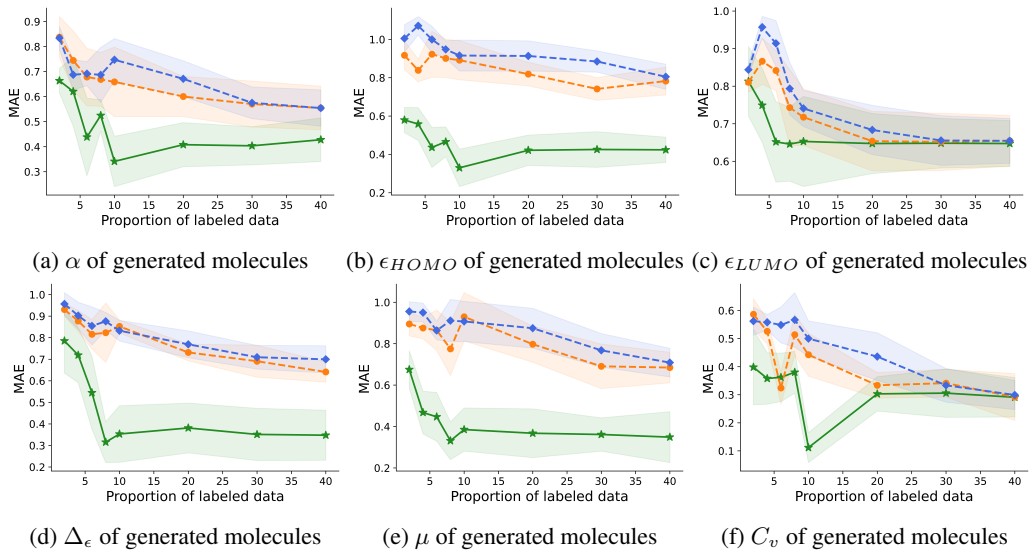

(a) $\alpha$ of generated molecules    (b) $\epsilon_{HOMO}$ of generated molecules    (c) $\epsilon_{LUMO}$ of generated molecules

(d) $\Delta_\epsilon$ of generated molecules    (e) $\mu$ of generated molecules    (f) $C_v$ of generated molecules

Figure 3: Evaluate controllability on Molecule dataset according to different proportions of paired data. Green solid line corresponds to Text2Data and two dashed lines are baseline comparisons, in which blue line is EDM and orange line is EDM-finetune. Properties of generated molecules are predicted by classifier $\phi_c$. MAE is computed between properties of generated molecules and intended properties. Lower MAE indicates better performance.

## 5.5 COMPARISONS ON CONTROLLABILITY

Figure 3 illustrates the MAE trend between the specific property of generated molecules and the intended one as the proportion of labeled training data rises. Text2Data achieves superior performance than EDM-finetune and EDM on all six properties. The results also indicate that certain properties, such as $\epsilon_{LUMO}$ and $Cv$, are more readily controllable. For these properties, the performance of the three models converges as the amount of labeled training data becomes sufficiently large.

We depict the molecules generated as the text descriptor for polarizability shifts from "very low" to "very high" in Figure 4. Polarizability indicates a molecule's inclination to form an electric dipole moment under an external electric field. As $\alpha$ values rise, we expect to see molecules with less symmetrical forms, as evidenced in Figure 4.

As suggested in Table 3, Text2Data also outperforms MDM-finetune and MDM in the controllable generation of motions from texts. While MDM-finetune is slightly better than Text2Data when the proportion of labeled training data is small-owing to milder catastrophic forgetting during finetuning with a smaller sample size-Text2Data consistently surpasses both MDM-finetune and MDM as the volume of labeled training data increases. Specifically, in this situation, Text2Data surpasses MDM-finetune and MDM in R Precision with average margins of 2.31% and 5.57%, respectively, and

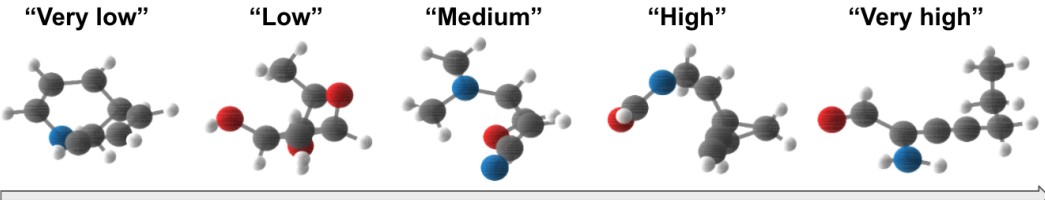

Figure 4: Visualization of generated molecules when the polarizability increases from "very low" to "very high".

in Multimodal Distance with average margins of 0.93% and 3.30%, respectively. The results also indicate that an increase in labeled training data enhances the performance of controllability.

Table 3: Evaluate controllability on HumanML3D dataset by R Precision and Multimodal Distance according to different proportions of paired data.

| Proportion (%) | R Precision ↑ | | | Multimodal Dist. ↓ | | |
|---|---|---|---|---|---|---|
| | Text2Data | MDM-finetune | MDM | Text2Data | MDM-finetune | MDM |
| 2 | 0.34±0.01 | **0.37±0.01** | 0.31±0.01 | 6.48±0.06 | **6.19±0.05** | 6.67±0.02 |
| 4 | 0.39±0.01 | **0.42±0.01** | 0.38±0.01 | 5.99±0.05 | **5.83±0.04** | 6.01±0.04 |
| 6 | 0.43±0.01 | **0.43±0.01** | 0.40±0.02 | 5.85±0.06 | **5.78±0.05** | 6.01±0.06 |
| 8 | **0.44±0.01** | 0.43±0.01 | 0.42±0.01 | **5.65±0.04** | 5.75±0.05 | 5.90±0.04 |
| 10 | **0.45±0.01** | 0.45±0.01 | 0.44±0.01 | **5.74±0.07** | 5.76±0.07 | 5.84±0.06 |
| 20 | **0.47±0.01** | 0.47±0.01 | 0.45±0.01 | **5.61±0.04** | 5.68±0.11 | 5.73±0.14 |
| 30 | **0.48±0.01** | 0.47±0.01 | 0.45±0.11 | **5.61±0.05** | 5.66±0.06 | 5.80±0.09 |
| 40 | **0.49±0.01** | 0.46±0.01 | 0.45±0.01 | **5.61±0.05** | 5.63±0.09 | 5.90±0.04 |

Table 4: Evaluate controllability on time series by MAE on testing set, according to different proportions of paired data. Lower MAE indicates better performance.

| Proportion (%) | Frequency ($\times 10^{-1}$) | | | Skewness | | | Mean ($\times 10^{-2}$) | | |
|---|---|---|---|---|---|---|---|---|---|
| | Text2Data | DiffTS-finetune | DiffTS | Text2Data | DiffTS-finetune | DiffTS | Text2Data | DiffTS-finetune | DiffTS |
| 2 | **2.59±0.20** | 2.62±0.20 | 2.60±0.17 | **1.68±0.20** | 2.34±0.30 | 1.84±0.22 | **0.63±0.39** | 0.63±0.39 | 0.79±0.41 |
| 4 | **2.55±0.18** | 2.59±0.19 | 2.59±0.19 | **1.63±0.14** | 1.77±0.29 | 2.80±0.28 | **0.60±0.40** | 0.61±0.39 | 0.73±0.40 |
| 6 | **2.52±0.18** | 2.57±0.19 | 2.57±0.19 | **1.00±0.16** | 1.85±0.22 | 1.81±0.21 | **0.56±0.38** | 0.58±0.38 | 0.71±0.36 |
| 8 | **2.54±0.18** | 2.56±0.19 | 2.57±0.18 | **1.10±0.18** | 1.56±0.18 | 1.78±0.09 | **0.57±0.38** | 0.63±0.40 | 0.62±0.36 |
| 10 | **2.54±0.19** | 2.57±0.18 | 2.55±0.20 | **0.87±0.13** | 1.20±0.17 | 1.12±0.11 | **0.55±0.37** | 0.57±0.36 | 0.62±0.39 |
| 20 | **2.54±0.18** | 2.55±0.18 | 2.58±0.21 | **1.05±0.15** | 1.06±0.12 | 1.26±0.14 | **0.55±0.40** | 0.57±0.37 | 0.65±0.38 |
| 30 | **2.53±0.18** | 2.56±0.17 | 2.56±0.18 | **1.03±0.12** | 1.16±0.25 | 1.71±0.23 | **0.51±0.33** | 0.53±0.33 | 0.59±0.34 |
| 40 | **2.53±0.18** | 2.55±0.18 | 2.55±0.19 | **1.03±0.11** | 1.15±0.24 | 1.19±0.18 | **0.51±0.33** | 0.57±0.32 | 0.57±0.36 |

We evaluate controllability of Text2Data, along with its baseline comparisons, by utilizing MAE to measure the congruence between the property of generated data and the intended one within the Time Series dataset. As indicated in Table 4, Text2Data consistently excels over DiffTS-finetune and DiffTS across all three properties assessed during the study. Results of another three properties are in Appendix Table 6, which leads to similar conclusion. Specifically, Text2Data and DiffTS-finetune show a marked improvement over DiffTS in controlling frequency, variance, and skewness. They also exhibit a slight edge in controlling mean, number of peaks, and linearity. The enhanced performance of Text2Data correlates with its proficiency in alleviating the issue of catastrophic forgetting while maintaining a pursuit of controllability.

## 6 CONCLUSION

In this paper, we propose a novel constraint optimization-based framework to improve the quality and controllability of text-to-data generation for various modalities in low-resource scenarios using diffusion models. We first identify three common practices to handle low-resource training along with their challenges. Subsequently, we introduce Text2Data, which employs unlabeled data to capture the prevailing data distribution using an unsupervised diffusion model. It is then finetunes on text-labeled data, employing a novel constraint optimization-based learning objective to ensure controllability while reducing catastrophic forgetting. Comprehensive experiments are conducted on real-world datasets and our model shows consistently superior performance to recent baselines.

While Text2Data is presented as a diffusion-based framework in this article, it can be seamlessly adapted to other generative models such as generative adversarial networks. The code has been attached to the supplemental materials to reproduce the results.

## 7 ETHICAL STATEMENT

We develop our method from publicly available QM9 (Ramakrishnan et al., 2014) and HumanML3D (Guo et al., 2022a) datasets, and stock data from public Yahoo Finance[2]. It is important to note that, like other text-to-data models, our implementation will likely reflect the socio-economic and entity biases inherent in datasets that we use. Additionally, although our method is designed for controllable data generation from text, we are not able to control the prompt that user inputs, which may contain improper contents.

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

## A  PROOF OF THEOREM 1

First, we define sub-exponential random variable and then and elucidate its relationship with sub-Gaussian random variable.

**Definition 2** (Sub-exponential random variable). The random variable $X$ with mean $\mu$ is sub-exponential with parameters $(v, b)$ if for $\forall \lambda < \frac{1}{b}$, $\mathbb{E}_X[\exp\{\lambda(X - \mu)\}] \leq \exp(\frac{v^2 \lambda^2}{2})$.

**Lemma 1.** The square of a zero-mean sub-Gaussian random variable with parameter $\sigma^2$ is a sub-exponential random variable with parameter $(4\sqrt{2}\sigma^2, 4\sigma^2)$ (Honorio & Jaakkola, 2014).

*Proof.* Given that $\boldsymbol{\epsilon}_\theta(\mathbf{x}^{(t)}, t)$ and $\boldsymbol{\epsilon}_\theta(\mathbf{x}^{(t)}, \mathbf{c}, t)$ are sub-Gaussian random variables parameterized by $\sigma^2$, and $\boldsymbol{\epsilon}$ is the multivariate standard Gaussian random variable, then their subtraction, $\boldsymbol{\epsilon}_\theta(\mathbf{x}^{(t)}, t) - \boldsymbol{\epsilon}$ and $\boldsymbol{\epsilon}_\theta(\mathbf{x}^{(t)}, \mathbf{c}, t) - \boldsymbol{\epsilon}$, are still sub-Gaussian random variables parameterized by $\sigma^2 + 1$. Then $\hat{\mathcal{L}}_1(\theta) = \mathbb{E}_{\mathbf{x} \sim \hat{p}_\mathcal{D}(\mathbf{x}), t}[||\boldsymbol{\epsilon}_\theta(\mathbf{x}^{(t)}, t) - \boldsymbol{\epsilon}||^2]$ and $\hat{\mathcal{L}}_2(\theta) = \mathbb{E}_{\mathbf{x}, \mathbf{c} \sim \hat{p}_{\mathcal{D}_p}(\mathbf{x}, \mathbf{c}), t}||\boldsymbol{\epsilon}_\theta(\mathbf{x}^{(t)}, \mathbf{c}, t) - \boldsymbol{\epsilon}||^2$, the square of sub-Gaussian random variables, are sub-exponential random variables parameterized by $(4\sqrt{2}\tilde{\sigma}^2, 4\tilde{\sigma}^2)$, based on the Lemma above.

Then, based on Bernstein's inequality, we have:

$$p(|\mathcal{L}_2(\theta) - \hat{\mathcal{L}}_2(\theta)| > \epsilon) \leq 2\exp(-\frac{N_p \epsilon^2}{8\sqrt{2}\tilde{\sigma}^2} \wedge \frac{N_p \epsilon}{8\tilde{\sigma}^2}) \tag{7}$$

Based on the union bound inequility, we further have:

$$p(\sup_{\theta \in \Theta} |\mathcal{L}_2(\theta) - \hat{\mathcal{L}}_2(\theta)| > \epsilon) \leq \sum_{\theta \in \Theta} p(|\mathcal{L}_2(\theta) - \hat{\mathcal{L}}_2(\theta)| > \epsilon)$$

$$\leq 2|\Theta| \exp(-\frac{N_p \epsilon^2}{8\sqrt{2}\tilde{\sigma}^2} \wedge \frac{N_p \epsilon}{8\tilde{\sigma}^2}) \tag{8}$$

Following the same way, we have:

$$p(\sup_{\theta \in \Theta} |\mathcal{L}'_1(\theta) - \hat{\mathcal{L}}'_1(\theta)| > \epsilon) \leq 2|\Theta| \exp(-\frac{N_p \epsilon^2}{8\sqrt{2}\tilde{\sigma}^2} \wedge \frac{N_p \epsilon}{8\tilde{\sigma}^2}) \tag{9}$$

$$p(\sup_{\theta \in \Theta} |\mathcal{L}_1(\theta) - \hat{\mathcal{L}}_1(\theta)| > \epsilon) \leq 2|\Theta| \exp(-\frac{N \epsilon^2}{8\sqrt{2}\tilde{\sigma}^2} \wedge \frac{N \epsilon}{8\tilde{\sigma}^2}) \tag{10}$$

Let the probability on the RHS of Eq. 8, Eq. 9 and Eq. 10 be $\delta$, then we compute $\epsilon$ and plug in into LHS, then with the probability of $1 - \delta$ we have:

$$\sup_{\theta \in \Theta} |\mathcal{L}_2(\theta) - \hat{\mathcal{L}}_2(\theta)| \leq \epsilon_{N_p} \tag{11}$$

$$\sup_{\theta \in \Theta} |\mathcal{L}'_1(\theta) - \hat{\mathcal{L}}'_1(\theta)| \leq \epsilon_{N_p} \tag{12}$$

$$\sup_{\theta \in \Theta} |\mathcal{L}_1(\theta) - \hat{\mathcal{L}}_1(\theta)| \leq \epsilon_N \tag{13}$$

where $\epsilon_N = \sqrt{8\sqrt{2}\tilde{\sigma}^2} \cdot \sqrt{\frac{\log|\Theta| + \log\frac{2}{\delta}}{N}} \vee 8\tilde{\sigma}^2 \cdot \frac{\log|\Theta| + \log\frac{2}{\delta}}{N}$ and $\epsilon_{N_p} = \sqrt{8\sqrt{2}\tilde{\sigma}^2} \cdot \sqrt{\frac{\log|\Theta| + \log\frac{2}{\delta}}{N_p}} \vee 8\tilde{\sigma}^2 \cdot \frac{\log|\Theta| + \log\frac{2}{\delta}}{N_p}$ are used to simplify the notation.

Let $\epsilon = \epsilon_N + \epsilon_{N_p}$. From Eq. 13, we have:

$$|\hat{\xi} - \xi| \leq \epsilon_N \tag{14}$$

$$\Longrightarrow \xi \leq \hat{\xi} + \epsilon_N \tag{15}$$

$$\Longrightarrow \xi + \epsilon_{N_p} \leq \hat{\xi} + \epsilon_{N_p} + \epsilon_N \tag{16}$$

Based on Eq. 12, similarly, we have $\hat{\mathcal{L}}_1'(\theta) \leq \mathcal{L}_1'(\theta) + \epsilon_{N_p}$. Since $\xi = \inf_{\theta \in \Theta} \mathcal{L}_1(\theta)$, then for $\forall \theta^*$ s.t. $\mathcal{L}_1'(\theta^*) = \xi$, according to Eq. 16, we can obtain:

$$\hat{\mathcal{L}}_1'(\theta) \leq \mathcal{L}_1'(\theta) + \epsilon_{N_p} \leq \xi + \epsilon_{N_p} \leq \hat{\xi} + \epsilon_{N_p} + \epsilon_N. \tag{17}$$

Let $\hat{\theta}^*$ be the solution of Eq. 6 and $\theta^*$ be the solution of Eq. 5. Additionally, let $\Theta^* = \{\theta : \mathcal{L}_1'(\theta) \leq \xi\}$, and $\hat{\Theta}^* = \{\theta : \hat{\mathcal{L}}_1'(\theta; \mathbf{x}) \leq \hat{\xi} + \epsilon\}$. We state that $\Theta^* \subseteq \hat{\Theta}^*$ based on Eq. 17.

Next, we prove that $\hat{\theta}^*$ competes well with $\theta^*$ on $\hat{\mathcal{L}}_2(\theta; \mathbf{x}, \mathbf{c})$ and $\mathcal{L}_2(\theta)$:

$$
\begin{aligned}
\mathcal{L}_2(\hat{\theta}^*) \leq & \hat{\mathcal{L}}_2(\hat{\theta}^*; \mathbf{x}, \mathbf{c}) + \epsilon_{N_p} && \text{based on Eq. 11} \\
= & \min_{\theta \in \hat{\Theta}^*} \hat{\mathcal{L}}_2(\theta; \mathbf{x}, \mathbf{c}) + \epsilon_{N_p} && \hat{\theta} \text{ is the solution of Eq. 6} \\
\leq & \min_{\theta \in \Theta^*} \hat{\mathcal{L}}_2(\theta; \mathbf{x}, \mathbf{c}) + \epsilon_{N_p} && \Theta^* \subseteq \hat{\Theta}^* \\
\leq & \min_{\theta \in \Theta^*} \mathcal{L}_2(\theta) + 2\epsilon_{N_p} && \text{based on Eq. 11} \\
= & \mathcal{L}_2(\theta^*) + 2\epsilon_{N_p} && (18)
\end{aligned}
$$

Next, we prove that $\hat{\theta}^*$ does not violate constraint too much:

$$
\begin{aligned}
\mathcal{L}_1'(\hat{\theta}^*) \leq & \hat{\mathcal{L}}_1'(\hat{\theta}^*; \mathbf{x}) && \text{based on Eq. 12} \\
\leq & \hat{\xi} + \epsilon + \epsilon_{N_p} && \text{based on definition of } \hat{\Theta}^* \\
= & \hat{\xi} + 2\epsilon_{N_p} + \epsilon_N && \epsilon = \epsilon_{N_P} + \epsilon_N \\
\leq & \xi + 2\epsilon_{N_p} + 2\epsilon_N && \text{based on Eq. 16} && (19)
\end{aligned}
$$

$\square$

## B  MORE DETAILS REGARDING DATASET

### B.1  STOCKS EMPLOYED TO FORM TIME SERIES DATA

We construct the dataset by assembling 24 stocks from Yahoo Finance during their IPO date to July 8, 2023, including Ethereum USD, NVIDIA Corporation, AT&T, Accenture plc, The Boeing Company, The Coca-Cola Company, Simon Property Group Inc., NIKE Inc., Sony Group Corporation, Chegg Inc., UnitedHealth Group Incorporated, General Motors Company, Russell 2000, JPMorgan Chase & Co., Salesforce Inc., Lockheed Martin Corporation, Walmart Inc., NASDAQ Composite, Shell plc, Pfizer Inc., Bitcoin USD, Apple Inc., Amazon.com Inc., Alphabet Inc.

### B.2  CONSTRUCTING TEXT DESCRIPTIONS

We construct the text descriptions of molecules by two templates: (1) exact description, e.g., "A molecule with the heat capacity of -0.11, the lomo of 0.87, the homo of -0.21, the polarizability of 0.95, the dipole moment of -1.61 and the energy gap between homo and lumo as 0.94.", and (2) general description, e.g., "A molecule with a high homo value, a very low heat capacity, a medium polarizability, a high energy difference between homo and lomo, a very high lomo value and a high dipole moment.". These text descriptions are then refined by GPT 3.5 and the order of different properties is shuffled. We form the text description for time series data using three templates: (a) exact description, e.g., "A time series with the frequency of 0.017, the mean of 3.12e-05, 19 peaks, the variance of 1.18e-11, the linear trend of 0.12 and the skewness of -6.15."; (b) general description, e.g., "A time series with large average, medium frequency, nearly equal large and small values, medium negative linearity, a few peaks and large variance" and (c) description of trend, e.g., "A time series that first stays stable, then increases with the slope of 1.25". Then those descriptions are further refined by GPT-3.5.

## C  IMPLEMENTATION DETAILS

### C.1  DETAILS OF TRAINING PROCESS

We leverage a cross-attention layer to enhance the alignment between data and text embeddings (Ma et al., 2023). Cross-attention is employed to enforce the alignment between data and text descriptions. Specifically, we maximize the cosine-similarity between text embeddings obtained from the

LLMs encoder and data embeddings diffused from original data. Baselines models are trained for 3,000 epoches. Their finetuned version and Text2Data are both pretrained for 1,000 epoches and finetuned for another 2,000 epoches. All experiments are conduced on A100 hardware.

### C.2 Score functions

For time series generation, we use the same score function as in Rasul et al. (2021) that consists of conditional 1-dim dilated ConvNets with residual connections adapted from the WaveNet (Oord et al., 2016) and Diffwave (Kong et al., 2020). For molecule generation, we employ the same score function from EDM Hoogeboom et al. (2022) which is a Multilayer Perceptron. For motion generation, we also leverage the same score function in Tevet et al. (2023) as a straightforward transformer.

### C.3 Training algorithm

The learning algorithm has been displayed in Alg. 1, where the model is firstly trained on unlabeled data, and then finetuned on labeled data under the framework of lexicographic optimization (Gong & Liu, 2021).

---

**Algorithm 1:** Learning phase

1  Alg. 1: General distribution learning with unlabeled data:
    **Input:** $\mathbf{x} \sim \hat{p}_{\mathcal{D}}(\mathbf{x})$
    **Input:** Pre-defined step size $\omega$
2  **while** *Converge* **do**
3      **for** *a batch of $B$ unlabeled data $\mathbf{x}_1, ..., \mathbf{x}_B$ sampled from $\hat{p}_{\mathcal{D}}(\mathbf{x})$* **do**
4          Initialize $t_i \sim Uniform(1, ..., T)$ for each $\mathbf{x}_i$, $i = 1, ..., B$, $\epsilon \sim \mathcal{N}(\mathbf{0}, \mathbf{I})$
5          Diffuse $\mathbf{x}_i$ to $\mathbf{x}_i^{(t_i)}$, $i = 1, ..., B$
6          Compute $\hat{\mathcal{L}}_1(\theta) = \frac{1}{B} \sum_{i=1}^{B} \|\epsilon_\theta(\mathbf{x}_i^{(t_i)}, t_i), -\epsilon\|^2$
7      Update $\theta$ by $\theta - \omega \cdot \nabla \hat{\mathcal{L}}_1(\theta)$
8  Alg. 2: Learning text-to-data generation by labeled data:
    **Input:** $\mathbf{x}, \mathbf{c} \sim \hat{p}_{\mathcal{D}_p}(\mathbf{x}, \mathbf{c})$
    **Input:** $\hat{\xi} = \inf_{\theta \in \hat{\Theta}} \hat{\mathcal{L}}_1(\theta)$ (i.e., minimum value obtained from Alg. 1)
    **Input:** Pre-defined positive hyperparameters $\alpha$, $\beta$, $\rho$ and step size $\omega'$
9  **while** *Converge* **do**
10     **for** *a batch of $B$ labeled data $(\mathbf{x}_1, \mathbf{c}_1), ..., (\mathbf{x}_B, \mathbf{c}_B)$ sampled from $\hat{p}_{\mathcal{D}_p}(\mathbf{x}, \mathbf{c})$* **do**
11         Initialize $t_i \sim Uniform(1, ..., T)$ for each $\mathbf{x}_i$, $i = 1, ..., B$, $\epsilon \sim \mathcal{N}(\mathbf{0}, \mathbf{I})$
12         Diffuse $\mathbf{x}_i$ to $\mathbf{x}_i^{(t_i)}$, $i = 1, ..., B$
13         Compute $\hat{\mathcal{L}}_2(\theta) = \frac{1}{B} \sum_{i=1}^{B} \|\epsilon_\theta(\mathbf{x}_i^{(t_i)}, \mathbf{c}_i, t_i), -\epsilon\|^2$
14         Compute $\hat{\mathcal{L}}_1'(\theta) = \frac{1}{B} \sum_{i=1}^{B} \|\epsilon_\theta(\mathbf{x}_i^{(t_i)}, t_i), -\epsilon\|^2$
15         Compute $\phi(\theta) = \min(\alpha(\hat{\mathcal{L}}_1'(\theta) - \rho \cdot \hat{\xi}), \beta\|\nabla\hat{\mathcal{L}}_1'(\theta)\|^2)$
16         Compute $\lambda = \max(\frac{\phi(\theta) - \nabla\hat{\mathcal{L}}_2(\theta)^T \nabla\hat{\mathcal{L}}_1'(\theta)}{\|\nabla\hat{\mathcal{L}}_1'(\theta)\|^2}, 0)$
17         Update $\theta$ by $\theta - \omega' \cdot (\nabla\hat{\mathcal{L}}_2(\theta) + \lambda\nabla\hat{\mathcal{L}}_1'(\theta))$

---

## D Additional results

The results of evaluating the generation quality of Text2Data and baseline comparisons on Molecule dataset based on molecular and atom stability are presented in Table 5.

Table 5: Evaluate generation quality on Molecule dataset by molecular and atom stability of generated molecules according to different proportions of paired data. Higher molecular stability and atom stability indicate better performance.

| Proportion (%) | Mol. Stability ↑ | | | Atom Stability ↑ | | |
|---|---|---|---|---|---|---|
| | Text2Data | EDM-finetune | EDM | Text2Data | EDM-finetune | EDM |
| 2 | **0.86±0.14** | 0.85±0.04 | 0.65±0.04 | **0.99±0.01** | **0.99±0.01** | 0.96±0.01 |
| 4 | **0.87±0.09** | 0.83±0.08 | 0.66±0.09 | **0.99±0.01** | 0.98±0.01 | 0.97±0.01 |
| 6 | **0.88±0.16** | 0.86±0.06 | 0.73±0.04 | **0.99±0.02** | **0.99±0.01** | 0.97±0.01 |
| 8 | **0.88±0.11** | 0.87±0.04 | 0.79±0.05 | **0.99±0.01** | **0.99±0.01** | 0.98±0.01 |
| 10 | **0.88±0.10** | 0.83±0.06 | 0.77±0.11 | **0.99±0.01** | 0.98±0.01 | 0.98±0.01 |
| 20 | **0.88±0.10** | 0.87±0.08 | 0.79±0.09 | **0.99±0.01** | 0.98±0.01 | 0.98±0.01 |
| 30 | **0.89±0.10** | 0.87±0.07 | 0.79±0.09 | **0.99±0.01** | 0.98±0.01 | 0.98±0.01 |
| 40 | **0.89±0.10** | 0.86±0.06 | 0.79±0.09 | **0.99±0.01** | 0.98±0.01 | 0.98±0.01 |

The results of evaluating the controllability of Text2Data and its baseline comparisons regarding variance, number of peaks and linearity of generated time series are in Table 6.

Table 6: Evaluate controllability on time series by MAE on testing set, according to different proportions of paired data. Lower MAE indicates better performance.

| Proportion (%) | Variance ($\times 10^{-5}$) | | | Number of peaks | | | Linearity | | |
|---|---|---|---|---|---|---|---|---|---|
| | Text2Data | DiffTS-finetune | DiffTS | Text2Data | DiffTS-finetune | DiffTS | Text2Data | DiffTS-finetune | DiffTS |
| 2 | **5.28±11.40** | 5.35±11.20 | 5.70±11.40 | **12.94±0.88** | 12.95±0.87 | 13.01±0.83 | 0.61±0.04 | **0.61±0.04** | 0.62±0.04 |
| 4 | **5.37±11.28** | 5.48±11.24 | 6.08±11.31 | 12.91±0.84 | **12.90±0.86** | 12.93±0.84 | **0.61±0.04** | 0.61±0.04 | 0.62±0.04 |
| 6 | **5.04±10.90** | 5.19±10.42 | 5.60±10.20 | **12.89±0.90** | 12.90±0.83 | 12.90±0.83 | **0.61±0.04** | 0.61±0.04 | 0.62±0.04 |
| 8 | **5.30±10.95** | 5.59±11.19 | 5.60±10.40 | **12.88±0.86** | 12.89±0.85 | 12.95±0.87 | **0.61±0.04** | 0.61±0.04 | 0.61±0.04 |
| 10 | **5.08±20.48** | 5.41±16.23 | 5.80±11.00 | **12.75±0.93** | 12.87±0.90 | 12.90±0.83 | **0.61±0.04** | 0.61±0.04 | 0.61±0.04 |
| 20 | **5.09±10.98** | 5.37±11.09 | 6.40±11.20 | **12.87±0.85** | 12.88±0.85 | 12.90±0.84 | **0.61±0.04** | 0.61±0.04 | 0.61±0.04 |
| 30 | **4.85±10.58** | 5.19±9.90 | 5.70±10.20 | **12.84±0.83** | 12.87±0.87 | 12.91±0.88 | **0.61±0.04** | 0.61±0.04 | 0.61±0.04 |
| 40 | **4.77±10.42** | 5.11±14.10 | 5.34±13.10 | **12.84±0.87** | 12.88±0.90 | 12.89±0.86 | **0.60±0.04** | 0.61±0.04 | 0.61±0.04 |

