# OpenReview forum: "Text2Data: Low-Resource Data Generation with Textual Control"
_ICLR.cc/2024/Conference — Submitted to ICLR 2024_

### Official Review · Reviewer_qy29 · 2023-10-28

**Soundness:** 3 good
**Presentation:** 3 good
**Contribution:** 2 fair
**Rating:** 6
**Confidence:** 2

**Summary:**

The paper presents an approach called Text2Data, which utilizes unlabeled data via an unsupervised diffusion model, then optimizes control to prevent forgetting. It efficiently utilizes both labeled and unlabeled data and employs a learning objective based on constraint optimization to prevent catastrophic forgetting during fine-tuning. The constrained optimization objective is based on lexicographic optimization. It minimizes the negative log-likelihood on labeled data, subject to the unsupervised loss on unlabeled data being less than some threshold. Experimental results show that Text2Data surpasses other benchmark methods in terms of generation quality and controllability.

**Strengths:**

Proposed method has two main stages:
1. Learn the data distribution from unlabeled data through an unsupervised diffusion model.
2. Finetune the model on limited labeled data using a novel constrained optimization objective. This aims to achieve controllability while preventing catastrophic forgetting of the distribution learned in stage 1.
The lexicographic approach involves first optimizing L2 as the primary objective without any constraints.
Then L'1 is optimized as a secondary objective by adding the constraint L'1 <= ξ, where ξ is determined by the minimum value of L'1 from the first optimization.  Thus it can balance labeled data fitting and remembering unlabeled data distribution.

**Weaknesses:**

1.There is no comparison to other techniques like data augmentation or transfer learning for low-resource text-to-data generation. Comparisons would be useful to know if Text2Data outperforms them.
2.The threshold ξ for the constrained optimization is set heuristically based on the minimum value of unsupervised loss L1.
3.Catastrophic forgetting is identified as an issue, but not experimentally verified by comparing with unconstrained finetuning. This would further prove the need for constrained optimization.

**Questions:**

My mainly technical concern on this paper is mainly on how to come up with thresholds for constrained optimization.

Another point is that the use of text annotation is perplexing. Given that prompts are either obtained from existing datasets or synthesized using templates and advanced language models like GPT-3, one might question: What is the fundamental difference between this approach and data augmentation (or using a classifier guidance)?

---

> ### Author Response · Authors · 2023-11-23
> **Response to reviewer qy29**
>
> **Comment 1:** There is no comparison to other techniques like data augmentation or transfer learning for low-resource text-to-data generation. Comparisons would be useful to know if Text2Data outperforms them.
>
> **Response 1:**
>
> * Data Augmentation: we employed GPT-4 to modify the textual descriptions to augment the text-data pairs (EDM-DA), and we directly performed classifier-free guidance diffusion on molecular generation using EDM. Below are the results that evaluate the controllability of EDM on properties of generated molecules (Alpha and HOMO).
>
> | %  | Alpha     |              |        |        | HOMO      |              |        |        |
> |----|-----------|--------------|--------|--------|-----------|--------------|--------|--------|
> |    | Text2Data | EDM-Finetune | EDM-DA | EDM    | Text2Data | EDM-Finetune | EDM-DA | EDM    |
> | 2  | 0.6636    | 0.8378       | 0.8002 | 0.8339 | 0.5792    | 0.9168       | 0.9534 | 1.0049 |
> | 4  | 0.6206    | 0.7446       | 0.6855 | 0.6874 | 0.5583    | 0.8384       | 0.9301 | 1.0708 |
> | 6  | 0.4384    | 0.6783       | 0.6736 | 0.6917 | 0.4364    | 0.9226       | 0.9005 | 0.9996 |
> | 8  | 0.5247    | 0.6687       | 0.6600 | 0.6868 | 0.4658    | 0.9008       | 0.8816 | 0.9477 |
> | 10 | 0.3412    | 0.6588       | 0.6128 | 0.7473 | 0.3299    | 0.8915       | 0.8825 | 0.9156 |
>
> Based on the results above, data augmentation-based approach slightly improves performance of directly finetuning the model. This is potentially due to the improvement of generalization of text descriptions.
>
> * We have already performed transfer learning based approach in our paper. For instance, EDM-Finetune represents the model that EDM is pretrained on a large amount of molecular structures, and then finetuned on a small amount of text-molecule pairs. Based on the results above, EDM-Finetune has much worse performance compared with the proposed approach. This is expected due to the potential catastrophic forgetting of transfer learning-based approach.
>
> **Comment 2:** The threshold $\xi$ for the constrained optimization is set heuristically based on the minimum value of unsupervised loss L1.
>
> **Response 2:**
>
> * The threshold does not have to be set strictly based on the minimum value of unsupervised loss L1, but can be further relaxed as long as it lies in the confidence bound of the minimum value of unsupervised loss L1.
>
> * To guide the relaxation of the threshold, we have derived a generalization bound as shown in Theorem 1. In Theorem 1, we have proved that the threshold can be further relaxed as long as it lies in the summation of empirical minimum value of unsupervised loss L1 and the confidence bound $\epsilon=\epsilon_N+\epsilon_{N_p}$. In practice, we multiply a hyperparameter before the threshold to relax it. We have highlighted relevant part in red in the main paper.
>
> **Comment 3:** Catastrophic forgetting is identified as an issue, but not experimentally verified by comparing with unconstrained finetuning. This would further prove the need for constrained optimization.
>
> **Response 3:**
>
> We have compared with unconstrained finetuning.  For instance, as shown in the table below, EDM-Finetune represents the model that EDM is pretrained on a large amount of molecular structures, and then finetuned on a small amount of text-molecule pairs without applying any constraints. Based on the results above, EDM-Finetune has much worse performance compared with the proposed approach. This is expected due to the potential catastrophic forgetting issue.
>
> | %  | Alpha     |              |        | HOMO      |              |        |
> |----|-----------|--------------|--------|-----------|--------------|--------|
> |    | Text2Data | EDM-Finetune | EDM    | Text2Data | EDM-Finetune | EDM    |
> | 2  | 0.6636    | 0.8378       | 0.8339 | 0.5792    | 0.9168       | 1.0049 |
> | 4  | 0.6206    | 0.7446       | 0.6874 | 0.5583    | 0.8384       | 1.0708 |
> | 6  | 0.4384    | 0.6783       | 0.6917 | 0.4364    | 0.9226       | 0.9996 |
> | 8  | 0.5247    | 0.6687       | 0.6868 | 0.4658    | 0.9008       | 0.9477 |
> | 10 | 0.3412    | 0.6588       | 0.7473 | 0.3299    | 0.8915       | 0.9156 |

---

### Official Review · Reviewer_m4kw · 2023-10-30

**Soundness:** 4 excellent
**Presentation:** 3 good
**Contribution:** 2 fair
**Rating:** 3
**Confidence:** 4

**Summary:**

The paper proposes a framework for data generation with controlling signal (e.g., text) where the controlling signal data is limited. To solve the data-scale challenge, the paper proposes to firstly train an unconditional generative model and then finetune the unconditional model with the limited paired dataset with controlling signal. To prevent the model from catastrophic forgetting, the paper proposes a constraint objective to keep the finetuned parameter space close to that of pretrained parameter.

**Strengths:**

1. The challenge addressed in this paper is meaningful. The paper is well-written and easy to read. The motivation of data-scale challenge and the solution is convincing.
2. The proposed method is verified on three different low-resource tasks to demonstrate the effectiveness.

**Weaknesses:**

1. As for the method, the proposed method is relevant to the classifier-free guidance, which both involves a conditional model and an unconditional model (and the two models share the parameters). The difference is that the proposed method: finetune the unconditional model while adding constraint for catastrophic forgetting and the classifier-free guidance: guidance the denoising direction (in diffusion model) of conditional model with the help of unconditional model. I think the discussion of why the proposed method is better than the classifier-free guidance is necessary.

2. As for the related work, the paper should discuss the related work about transfer learning, where catastrophic forgetting is an important topic. Since one of the contributions of the paper is preventing catastrophic forgetting. I think the paper should compare the proposed method with other works about catastrophic forgetting.

3. As for the experiments, I understand the proposed method is proposed for low-resource condition. However, the proposed method should be verified on popular tasks such as image generation (maybe the author can show that the proposed method can achieve a high performance with only 5% data compared with the conventional image generation methods). Currently the 3 task in paper is not quite convincing. And the time series task even has no previous published baselines (the baseline model is also proposed by the author).

**Questions:**

1. How the constraint in Eq. 6 is achieved? Whether it is applied with an auxiliary loss function? If so, which kinds of loss function?

2. What is the text description in  time series task? Can you give some examples? I am also curious that why the time series task (seems) does not conduct prediction task? Or can you give a more detailed description about the time series task?

---

> ### Author Response · Authors · 2023-11-23
> **Response to reviewer m4kw**
>
> **Comment 1:** As for the method, the proposed method is relevant to the classifier-free guidance, which both involves a conditional model and an unconditional model (and the two models share the parameters). The difference is that the proposed method: finetune the unconditional model while adding constraint for catastrophic forgetting and the classifier-free guidance: guidance the denoising direction (in diffusion model) of conditional model with the help of unconditional model. I think the discussion of why the proposed method is better than the classifier-free guidance is necessary.
>
> **Response 1:** We target the situation while labeled data is very limited. Directly learning a conditional model on such limited data may not have satisfactory generalizability and suffer from the limited diversity of generated data. Additionally, as there are too many parameters in the diffusion model (e.g., 14 million for Motion Diffusion Model), the lack of labeled data may cause issues such as overfitting. By contrast, our model is first pre-trained on unlabeled data, which usually ensures enough sample size to avoid overfitting.
>
> **Comment 2:** As for the related work, the paper should discuss the related work about transfer learning, where catastrophic forgetting is an important topic. Since one of the contributions of the paper is preventing catastrophic forgetting. I think the paper should compare the proposed method with other works about catastrophic forgetting.
>
> **Response 2:**
>
> * We have discussed representative works of transfer learning and its potential drawbacks in causing catastrophic forgetting. We have highlighted this part in red for easier reference:
>
> “Tu et al. (2019) proposes to learn a mapping between source and target linguistic symbols and employs transfer learning to transfer knowledge from a high-resource language to low-resource language.”
>
> “Nevertheless, all those strategies have their own limitations, such as high computational complexity for data augmentation, difficulty in maintaining the correct interpretation of text when leveraging unlabeled data during semi-supervised learning, and the potential catastrophic forgetting issues in transfer learning.”
>
> * We have also conducted additional experiments for data augmentation, another approach to overcome catastrophic forgetting.
>
> Specifically, we employed GPT-4 to modify the textual descriptions to augment the text-data pairs (EDM-DA), and we directly performed classifier-free guidance diffusion on molecular generation using EDM. Below are the results that evaluate the controllability of EDM on properties of generated molecules (Alpha and HOMO).
>
> | %  | Alpha     |              |        |        | HOMO      |              |        |        |
> |----|-----------|--------------|--------|--------|-----------|--------------|--------|--------|
> |    | Text2Data | EDM-Finetune | EDM-DA | EDM    | Text2Data | EDM-Finetune | EDM-DA | EDM    |
> | 2  | 0.6636    | 0.8378       | 0.8002 | 0.8339 | 0.5792    | 0.9168       | 0.9534 | 1.0049 |
> | 4  | 0.6206    | 0.7446       | 0.6855 | 0.6874 | 0.5583    | 0.8384       | 0.9301 | 1.0708 |
> | 6  | 0.4384    | 0.6783       | 0.6736 | 0.6917 | 0.4364    | 0.9226       | 0.9005 | 0.9996 |
> | 8  | 0.5247    | 0.6687       | 0.6600 | 0.6868 | 0.4658    | 0.9008       | 0.8816 | 0.9477 |
> | 10 | 0.3412    | 0.6588       | 0.6128 | 0.7473 | 0.3299    | 0.8915       | 0.8825 | 0.9156 |
>
> Based on the results above, data augmentation-based approach slightly improves performance of directly finetuning the model. This is potentially due to the improvement of generalization of text descriptions.
>
> **Comment 3:** The proposed method should be verified on popular tasks such as image generation. Currently the 3 task in paper is not quite convincing.
>
> **Response 3:** We did not apply our approach to domains such as image generation because those domains usually have huge labeled dataset (e.g., ImageNet, CelebFaces, COCO) that can be leveraged to directly train the diffusion model conditional on labels. Instead, we focus on modalities such as time series, molecules or motions that lack textual descriptions due to costly annotations or intricate data structures.

---

### Official Review · Reviewer_Yf2d · 2023-11-05

**Soundness:** 3 good
**Presentation:** 2 fair
**Contribution:** 3 good
**Rating:** 5
**Confidence:** 3

**Summary:**

This work studies data generation conditioned on textual control under low-resource scenarios. The key idea is to conduct large-scale pre-training to learn the data marginal distribution; Then solving the conditioned generation while mitigating the catastrophic forgetting issue.

**Strengths:**

The authors tackle the catastrophic forgetting issue of the pre-training then fine-tuning strategy via solving a constrained optimization problem. The authors provide a textual description on how they come to the objective function, from equation 4 to 5 and then to 6; A proof is proved to show that the relaxation from 5 to 6 is fine given some assumptions.



The proposed baselines and the evaluation metrics all make sense. The authors manage to show that their constrained optimization can provide benefits. Actually, the proposed method shows benefits in the evaluations the authors have conducted.

**Weaknesses:**

Details/Clarity are missing:

— The authors claim that it is “usually easy to obtain by either manual collection of simulation” to get the unlabeled data. What’s the estimated manual data collecting effort and procedure? How do the authors design a system to do simulation? If data synthesis is based on an existing generative model, to train such a high-quality generative model already requires a lot of data.

— The authors claim they use a cross-attention layer, but some implementation details are missing.

— What’s the pre-training data? Do the authors use the same dataset of the downstream task for pre-training? By reading the 2nd paragraph of page 6, here’s my understanding: The authors identified N datasets with textual description. The authors create the synthesized ‘low-resource’ scenario by sampling varying proportions of the dataset. Pre-training is done on the full dataset for each concrete problem.

Some more questions/comments:
-- The authors should have a baseline that trained on the 100% of the labeled data.

-- Please include proportions that are larger than 40 as reference.

-- What if the model is not in-domain pre-trained? For example, a generative model really pre-trained on large-scale data, but does not use the downstream dataset.

-- The authors aim to solve the problem under a low-resource scenario. However, the proposed methods do not always show benefits when using 2%, 4% and 6% data, according to Table one and two.

**Questions:**

Questions are posted in Weaknesses section

---

> ### Author Response · Authors · 2023-11-23
> **Response to reviewer Yf2d**
>
> **Comment 1:** What’s the estimated manual data collecting effort and procedure? How do the authors design a system to do simulation? If data synthesis is based on an existing generative model, to train such a high-quality generative model already requires a lot of data.
>
> **Response 1:**
>
> * Manual collection can be the existing unlabeled datasets such as QM9, ZINC, etc. Those datasets usually contains a huge amount of unlabeled samples to pretrain our model. For instance, QM9 has 134k stable organic molecules, and ZINC has 250k commercially available drug-like chemical compounds.
>
> * Simulation can be, for example, simulating time series data from a statistical model, such as mixture autoregressive model [1], autoregressive moving-average model [2] etc. From those statistical models we can simulate as many as samples of time series, but they are still unlabeled.
>
> [1] Kang et al., GeneRAting TIme Series with diverse and controllable characteristics.
>
> [2] Benjamin et al., Generalized autoregressive moving average models.
>
> **Comment 2:** The authors claim they use a cross-attention layer, but some implementation details are missing.
>
> **Response 2:**
>
> We use the cross-attention to enforce the alignment between data and text descriptions. Specifically, we maximize the cosine-similarity between text embeddings obtained from the LLMs encoder and data embeddings diffused from original data. We have added the explanation to Appendix C.1 and the relevant part has been highlighted in red.
>
> **Comment 3:** Pre-training is done on the full training dataset, that involves both unlabeled and labeled data. Labeled data is treated as unlabeled without using their labels in the pre-training process. Finetuning relies on only those small proportion of labeled data in the training set.
>
> **Response 3:** Pre-training is done on the full training dataset, that involves both unlabeled and labeled data. Labeled data is treated as unlabeled without using their labels in the pre-training process. Finetuning relies on only those small proportion of labeled data in the training set.
>
> **Comment 4:** The authors should have a baseline that trained on the 100% of the labeled data. -- Please include proportions that are larger than 40 as reference.
>
> **Respone 4:** We have added experiments for up to 100% of labeled data (e.g., 60%, 80%, 100%), the results of which are as below. The experiments are performed on QM9 dataset by leveraging EDM baseline.
>
> | %   | Alpha     |              |        | HOMO      |              |        |
> |-----|-----------|--------------|--------|-----------|--------------|--------|
> |     | Text2Data | EDM-Finetune | EDM    | Text2Data | EDM-Finetune | EDM    |
> | 2   | 0.6636    | 0.8378       | 0.8339 | 0.5792    | 0.9168       | 1.0049 |
> | 4   | 0.6206    | 0.7446       | 0.6874 | 0.5583    | 0.8384       | 1.0708 |
> | 6   | 0.4384    | 0.6783       | 0.6917 | 0.4364    | 0.9226       | 0.9996 |
> | 8   | 0.5247    | 0.6687       | 0.6868 | 0.4658    | 0.9008       | 0.9477 |
> | 10  | 0.3412    | 0.6588       | 0.7473 | 0.3299    | 0.8915       | 0.9156 |
> | 20  | 0.4079    | 0.6004       | 0.6711 | 0.4241    | 0.8188       | 0.9134 |
> | 40  | 0.4276    | 0.5546       | 0.5543 | 0.4238    | 0.7825       | 0.8050 |
> | 60  | 0.4100    | 0.5011       | 0.5151 | 0.4300    | 0.6055       | 0.5912 |
> | 80  | 0.4025    | 0.4050       | 0.4317 | 0.4217    | 0.5034       | 0.5120 |
> | 100 | 0.4006    | 0.4019       | 0.3994 | 0.4189    | 0.4295       | 0.4234 |
>
> Based on the results in the table above, both finetuned method without constraint (i.g., EDM-Finetune) and the direct conditional model (EDM) converge when the proportion goes up to 100%.

---

> ### Author Response · Authors · 2023-11-23
> **Response to reviewer Yf2d**
>
> **Comment 5:** What if the model is not in-domain pre-trained? For example, a generative model really pre-trained on large-scale data, but does not use the downstream dataset.
>
> **Response 5:** We have added experiments for our model to pre-train using just unlabeled data and finetune on labeled data (i.e., EDM-In).
>
> | %  | Alpha     |              |        |        | HOMO      |              |        |        |
> |----|-----------|--------------|--------|--------|-----------|--------------|--------|--------|
> |    | Text2Data | EDM-Finetune | EDM-In | EDM    | Text2Data | EDM-Finetune | EDM-In | EDM    |
> | 2  | 0.6636    | 0.8378       | 0.6678 | 0.8339 | 0.5792    | 0.9168       | 0.5840 | 1.0049 |
> | 4  | 0.6206    | 0.7446       | 0.6455 | 0.6874 | 0.5583    | 0.8384       | 0.5834 | 1.0708 |
> | 6  | 0.4384    | 0.6783       | 0.5571 | 0.6917 | 0.4364    | 0.9226       | 0.5024 | 0.9996 |
> | 8  | 0.5247    | 0.6687       | 0.5412 | 0.6868 | 0.4658    | 0.9008       | 0.4969 | 0.9477 |
> | 10 | 0.3412    | 0.6588       | 0.4556 | 0.7473 | 0.3299    | 0.8915       | 0.4310 | 0.9156 |
>
> The experiments are performed on molecular generation under the proportion of labeled data from 2% up to 10%. The results show that the performance of EDM-In is slightly worse than Text2Data, but much better than EDM-Finetune and EDN. This is expected since the only drawback of using only unlabeled data in pretraining is the reduction of sample size to learn the general data distribution. But from 2% to 10%, the loss of sample size is pretty small, which does not harm the performance too much.

---

### Official Review · Reviewer_L6cR · 2023-11-10

**Soundness:** 3 good
**Presentation:** 2 fair
**Contribution:** 2 fair
**Rating:** 3
**Confidence:** 2

**Summary:**

The authors propose a model for performing semi-supervised learning that first trains a model to maximise marginal likelihood
 and then fine tunes the same on labelled data by enforcing a constraint that the marginal likelihood of the fine-tuned model should not deviate much from the starting point. This is modelled as a constraint during optimisation.

The authors introduce a relaxation for the constraint and prove that under the relaxed constraint, the set of \theta that minimises the empirical loss contains the true \theta (the one that minimises the true loss)

**Strengths:**

The primary contribution of this paper is the constrained-optimisation objective  and the generalisation bound on the constrained optimisation problem. Since learning theory is not my area of expertise, it is difficult for me to evaluate the generalization bound contribution in terms of correctness and novelty.

**Weaknesses:**

The approach of optimising marginal likelihood on unlabelled data and conditional likelihood on labelled data is very common in semi-supervised learning literature [1]. The primary difference is that here the authors have also included the marginal likelihood on unlabelled data as a constraint. This would have been interesting if the authors had discussed how that constraint is used during training batch-by-batch. Unfortunately the algorithm for training is missing from the paper.

There are lots of other details missing from the paper as well. For instance, how is $\epsilon_\theta$ modelled? How can it be a function of x at one point and x and c at the other point. Perhaps the authors have assumed that a lot of stuff to be obvious.

There are no novel insights from the paper. It is not clear why a constrained optimisation objective is superior over other forms of semi-supervised learning. This weakens the paper a lot.

[1] Semi-Supervised Learning with Deep Generative Models" Authors: Diederik P. Kingma, Danilo J. Rezende, Shakir Mohamed, Max Welling

**Questions:**

Included in weaknesses section

---

> ### Author Response · Authors · 2023-11-23
> **Response to reviewer L6cR**
>
> **Comment 1:** The approach of optimising marginal likelihood on unlabelled data and conditional likelihood on labelled data is very common in semi-supervised learning literature. The primary difference is that here the authors have also included the marginal likelihood on unlabelled data as a constraint. This would have been interesting if the authors had discussed how that constraint is used during training batch-by-batch. Unfortunately the algorithm for training is missing from the paper.
>
> **Response 1:**
>
> * We train the model following a pretrain-finetune manner. We firstly pretrain the model on unlabeled data to capture the general data distribution $p_\theta(x)$, as usually people have enough unlabeled data. Then we finetune our model on (usually small amount of) labeled data while, based on Eq. (2), ensuring the finetuned distribution not far from the one learned during the pretrain process. This is not strictly semi-supervised learning as we do not infer labels of unlabeled data, but just leverage unlabeled data to learn a good distribution to initiate finetuning process.
>
> * We have added an algorithm of the training process in Appendix C.3 of revised paper.
>
> **Comment 2:** There are lots of other details missing from the paper as well. For instance, how is \epsilon_theta modelled? How can it be a function of x at one point and x and c at the other point. Perhaps the authors have assumed that a lot of stuff to be obvious.
>
> We just borrowed the score function of the traditional classifier-free diffusion guidance [1]. We have added the detailed derivation of the diffusion score in Appendix C.2:
>
> “For time series generation, we use the same score function as in [2] that consists of conditional 1-dim dilated ConvNets with residual connections adapted from the WaveNet [3] and Diffwave [4]. For molecule generation, we employ the same score function from EDM [5] which is a Multilayer Perceptron. For motion generation, we also leverage the same score function in [6] as a straightforward transformer.”
>
> We also highlighted this part in red for easier reference of the reviewer.
>
> [1] Ho and Salimans, Classifier-free Diffusion Guidance.
>
> [2] Rasul et al., Autoregressive denoising diffusion models for multivariate probabilistic time series forecasting.
>
> [3] Oord et al., Wavenet: A generative model for raw audio.
>
> [4] Kong et al., Diffwave: A versatile diffusion model for audio synthesis.
>
> [5] Hoogeboom et al., Equivariant diffusion for molecule generation in 3d.
>
> [6] Tevet et al., Human motion diffusion model.
>
> **Comment 3:** There are no novel insights from the paper. It is not clear why a constrained optimisation objective is superior over other forms of semi-supervised learning. This weakens the paper a lot.
>
> **Response 3:**
>
> * We have explained the drawbacks of semi-supervised learning approaches as follows:
>
> “For semi-supervised learning, text inherently carries nuances, ambiguities, and multiple meanings. Ensuring that the model maintains the correct interpretation when leveraging unlabeled data is not straightforward.”
>
> We have highlighted this part in our paper for easier reference.
>
> * Our approach is not a strictly semi-supervised learning-based approach as our approach leverages unlabeled data just to learn a general distribution of data, without inferring any of its labels. Therefore, this will not introduce any text-data ambiguity during the learning process. Then during finetuning, we uses (usually) a small amount of labeled data to finetune the model conditional on their text descriptions. This process will not cause any ambiguities either as we are using existing textual descriptions, not inferring their textual semantics.

---

> > ### Comment · Reviewer_L6cR · 2023-11-23
> >
> > > Our approach is not a strictly semi-supervised learning-based approach as our approach leverages unlabeled data just to learn a general distribution of data, without inferring any of its labels
> >
> > To confirm, does equation (6) only use labelled data?
> >
> > According to equation (2), the marginal p(x) is almost equal to the conditional p(x|c) where (x,c)~D_p as training proceeds. So, as training progresses, both L_1 and L_2' in equation (6) become almost equal. In other words L_2' doesn't serve as a regulariser if you use only labelled data in equation (6).

---

> > > ### Author Response · Authors · 2023-11-23
> > > **Response to reviewer L6cR**
> > >
> > > We sincerely appreciate the reviewer's response and comments, and giving us the opportunity to clarify our paper.
> > >
> > > * Yes, Eq. (6) only uses labelled data.
> > >
> > > * The conditional distribution (i.e., $p_{\theta}(x|c)$) should be close to the marginal distribution of data (i.e., $p_{\theta}(x)$) based on Eq. (2). We have to make sure the distribution (i.e., $p_{\mathcal D_p}(x|c)$) we learned via $\hat{\mathcal L_2}$ satisfy this constraint. Therefore, in Eq. (6), we add a constraint regarding $\hat{\mathcal L_1}$', which shares the same set of parameters as $\hat{\mathcal L_2}$. Note that the marginal distribution of the left hand side of Eq. (2) is not $p_{\mathcal D_p}(x)$ but $p_{\mathcal D}(x)$, where $\mathcal D_p\subset \mathcal D$. We use labelled data (i.e., $(x, c)\sim p_{\mathcal D_p}(x, c)$) to compute $\hat{\mathcal L_2}$, and we use the same set of $x$ to compute $\hat{\mathcal L_1}$' to mirror the learning objective to learn $p_{\mathcal D}(x)$ during pretraining (i.e., same learning objective as $\hat{\mathcal L_1}$ but with $x\sim p_{\mathcal D_p}(x)$). This can make sure the learned parameter during finetuning is not far from those learned during pretraining.

---

> ### Comment · Reviewer_L6cR · 2023-11-23
>
> Perhaps I am missing something, but if you minimise only $\hat{L}_2$ in equation (6), won't it automatically minimise $\hat{L}_1'$ In particular, if p(x|c) is very high for a particular x, doesn't it automatically imply that p(x) is automatically going to be very high.
>
> We don't need that extra term for  $\hat{L}_1'$ at all unless we compute it for unlabelled data for which we don't know $c$. Am I wrong here?
>
> This is why in semi-supervised learning, the marginal-loss term is always over the unlabelled data

---

### Meta-Review · Area_Chair_uaeb · 2023-12-06

**Metareview:**

The submission introduces Text2Data, a framework for data generation with controlled signals in low-resource settings. It begins with large-scale pre-training to learn the data's marginal distribution and then fine-tunes the model using limited paired data while addressing the challenge of catastrophic forgetting. This is achieved through a constraint that ensures the fine-tuned model remains close to the pretrained one.

The reviewers are lukewarm about the submission.
The authors prepared the rebuttal, but the reviewers' comments and concerns were only partially addressed.
The remaining concerns include limited scenarios (low-resource), limited effectiveness, and unaddressed questions.

**Justification For Why Not Higher Score:**

This submission has obtained weak support from the reviewers. While the authors put their efforts into the rebuttal, it seems not to be sufficiently convincing to the reviewers. This submission needs additional efforts to better pose the work.

**Justification For Why Not Lower Score:**

.

---

### Decision · Program_Chairs · 2024-01-16

Reject